# Serious Long-Term Effects of Head and Neck Cancer from the Survivors’ Point of View

**DOI:** 10.3390/healthcare11060906

**Published:** 2023-03-21

**Authors:** Katherine J. Taylor, Cecilie D. Amdal, Kristin Bjordal, Guro L. Astrup, Bente B. Herlofson, Fréderic Duprez, Ricardo R. Gama, Alexandre Jacinto, Eva Hammerlid, Melissa Scricciolo, Femke Jansen, Irma M. Verdonck-de Leeuw, Giuseppe Fanetti, Orlando Guntinas-Lichius, Johanna Inhestern, Tatiana Dragan, Alexander Fabian, Andreas Boehm, Ulrike Wöhner, Naomi Kiyota, Maximilian Krüger, Pierluigi Bonomo, Monica Pinto, Sandra Nuyts, Joaquim C. Silva, Carmen Stromberger, Francesco Tramacere, Ayman Bushnak, Pietro Perotti, Michaela Plath, Alberto Paderno, Noa Stempler, Maria Kouri, Susanne Singer

**Affiliations:** 1Institute of Medical Biostatistics, Epidemiology, and Informatics, University Medical Centre Mainz, 55131 Mainz, Germany; 2Department of Oncology, Oslo University Hospital, 0372 Oslo, Norway; 3Research Support Service, Oslo University Hospital 0372 Oslo, Norway; 4Faculty of Medicine, University of Oslo, 0372 Oslo, Norway; 5Faculty of Dentistry, University of Oslo, 0455 Oslo, Norway; 6Department of Otorhinolaryngology, Oslo University Hospital, 0372 Oslo, Norway; 7Department of Radiotherapy-Oncology, Faculty of Medicine and Health Sciences—Human Structure and Repair, Ghent University Hospital, Ghent University, 9000 Ghent, Belgium; 8Department of Head and Neck Surgery, Barretos Cancer Hospital, Barretos 14784-400, SP, Brazil; 9Department of Radiation Oncology, Barretos Cancer Hospital, Barretos 14784-400, SP, Brazil; 10Department of Otorhinolaryngology-Head and Neck Surgery, Institute of Clinical Sciences, Sahlgrenska Academy at University of Gothenburg, Sahlgrenska University Hospital, 41345 Gothenburg, Sweden; 11Department of Radiation Oncology, Ospedale dell’Angelo, 30174 Venice, Italy; 12Department Otolaryngology-Head and Neck Surgery, Amsterdam UMC Location Vrije Universiteit, De Boelelaan 1117, 1081 HZ Amsterdam, The Netherlands; 13Cancer Center Amsterdam, Treatment and Quality of Life, 1081 HV Amsterdam, The Netherlands; 14Department Clinical, Neuro and Developmental Psychology, Vrije Universiteit Amsterdam, Van der Boechorststraat 7-9, 1081 HV Amsterdam, The Netherlands; 15Division of Radiation Oncology, Centro di Riferimento Oncologico di Aviano (CRO) IRCCS, 33081 Aviano, Italy; 16Department of Otorhinolaryngology, Jena University Hospital, 07747 Jena, Germany; 17Department of Otorhinolaryngology, Oberhavelkliniken Hennigsdorf, 16761 Hennigsdorf, Germany; 18Head and Neck Unit, Department of Radiation Oncology, Institut Jules Bordet, Université Libre de Bruxelles, 1070 Brussels, Belgium; 19Department of Radiation Oncology, University Hospital Schleswig-Holstein, 24105 Kiel, Germany; 20Department of Otorhinolaryngology, St. Georg Hospital, 04129 Leipzig, Germany; 21Cancer Center, Kobe University Hospital, Kobe 650-0017, Japan; 22Department of Oral and Maxillofacial Surgery—Plastic Surgery, University Medical Centre Mainz, 55131 Mainz, Germany; 23Radiation Oncology, Azienda Ospedaliero-Universitaria Careggi, 50134 Florence, Italy; 24Rehabilitation Medicine Unit, Strategic Health Services Department, Istituto Nazionale Tumori—IRCCS—Fondazione G. Pascale, 80131 Naples, Italy; 25Laboratory of Experimental Radiotherapy, Department of Oncology, KU Leuven, 3000 Leuven, Belgium; 26Department of Radiation Oncology, Leuven Cancer Institute, University Hospitals Leuven, 3000 Leuven, Belgium; 27Department of Otolaryngology, Head and Neck Surgery, Instituto Português de Oncologia Francisco Gentil do Porto, 4200-072 Porto, Portugal; 28Department of Radiation Oncology, Charité-Universitätsmedizin Berlin, 13353 Berlin, Germany; 29Berlin Institute of Health, 10178 Berlin, Germany; 30Department of Radiation Oncology, Azienda Sanitaria Locale, 72100 Brindisi, Italy; 31Department of Otorhinolaryngology, University Hospital Gießen und Marburg, 35392 Giessen, Germany; 32Department of Otorhinolaryngology—Head and Neck Surgery, “S. Chiara” Hospital, Azienda Provinciale per i Servizi Sanitari (APSS), 38122 Trento, Italy; 33Department of Otorhinolaryngology, Head and Neck Surgery, University Hospital Heidelberg, 69120 Heidelberg, Germany; 34Department of Otorhinolaryngology—Head and Neck Surgery, ASST Spedali Civili of Brescia, University of Brescia, 25123 Brescia, Italy; 35Oral Medicine Unit, Sheba Medical Center, Tel Hashomer, Ramat Gan 5262000, Israel; 36Dental Oncology Unit, Department of Oral Medicine and Pathology and Hospital Dentistry, Dental School, National and Kapodistrian University of Athens, 11527 Athens, Greece

**Keywords:** head and neck cancer, patient-reported outcomes, survivor, quality of life, side effect

## Abstract

The long-term problems of head and neck cancer survivors (HNCS) are not well known. In a cross-sectional international study aimed at exploring the long-term quality of life in this population, 1114 HNCS were asked to state their two most serious long-term effects. A clinician recorded the responses during face-to-face appointments. A list of 15 example problems was provided, but a free text field was also available. A total of 1033 survivors responded to the question. The most frequent problems were ‘dry mouth’ (DM) (*n* = 476; 46%), ‘difficulty swallowing/eating’ (DSE) (*n* = 408; 40%), ‘hoarseness/difficulty speaking’ (HDS) (*n* = 169; 16%), and ‘pain in the head and neck’ (PHN) (*n* = 142; 14%). A total of 5% reported no problems. Logistic regression adjusted for age, gender, treatment, and tumor stage and site showed increased odds of reporting DM and DSE for chemo-radiotherapy (CRT) alone compared to surgery alone (odds ratio (OR): 4.7, 95% confidence interval (CI): 2.5–9.0; OR: 2.1, CI: 1.1–3.9), but decreased odds for HDS and PHN (OR: 0.3, CI: 0.1–0.6; OR: 0.2, CI: 0.1–0.5). Survivors with UICC stage IV at diagnosis compared to stage I had increased odds of reporting HDS (OR: 1.9, CI: 1.2–3.0). Laryngeal cancer survivors had reduced odds compared to oropharynx cancer survivors of reporting DM (OR: 0.4, CI: 0.3–0.6) but increased odds of HDS (OR: 7.2, CI: 4.3–12.3). This study provides evidence of the serious long-term problems among HNCS.

## 1. Introduction

Of the estimated nearly 19.3 million new cancer cases diagnosed worldwide in 2020, approximately 932,000 (~5%) were head and neck cancer (HNC) cases, making it the seventh most frequent diagnosis for that year. In terms of mortality, in 2020 around 467,000 deaths were attributed to HNC [1]. Five-year relative survival proportions vary considerably depending on the specific site and extent of the tumor at diagnosis, but reported examples include 66% across all HNC, 34% for hypopharynx, and up to 97% for lip cancers [2]. The disease and its treatment are usually accompanied by significant side effects. Health-related quality of life (HRQoL) research following an HNC diagnosis has shown that patients report a wide range of issues, including pain, hearing loss, financial burden, body image disturbance, breathing and eating problems, dry mouth, sticky saliva, and fatigue [3,4,5,6,7,8]. The HRQoL of long-term survivors of HNC has not been sufficiently researched, although there is patient-reported evidence that patients continue to suffer from speech problems, dry mouth, as well as eating and swallowing problems [9,10,11,12,13]. This evidence has been found using validated questionnaires, such as the EORTC quality of life core questionnaire and the corresponding head and neck module, which permit scores to be calculated across various symptom domains based on the patients’ responses, with high symptom scores indicating a high symptom burden [14,15]. Questionnaires simultaneously assessing a range of symptoms provide valuable information on the extent of symptoms and problems, but not necessarily which problem(s) the patient sees as his/her most pressing issue. For example, a study exploring methods to determine the minimal important difference and minimal important change in HRQoL scores found that some patients indicated that their quality of life in terms of swallowing ability had improved between two time points although their swallowing score indicated that their symptoms had worsened in the same time period [16]. In addition, an indication of what the patient feels is causing them the most distress could help direct intervention efforts to best help the patient. In a large study on the long-term effects of HNC and its treatment on patients, we included one question asking exactly this. This article reports the results of directly asking long-term survivors about their most serious current long-term effects.

## 2. Methods

The “Late Toxicity and Long-time Quality of Life in HNC Survivors” project (EORTC 1629) is an international cross-sectional study aimed at assessing the long-term quality of life and toxicities present in survivors of HNC. To participate, survivors had to be five years or more post-diagnosis. Additional inclusion criteria included being at least 18 years old at the time the invitation to participate was sent, being able to understand questionnaires in the local language, and having no diagnosis of lymphoma or skin cancer in the head and neck.

In the study’s case report form (CRF), a question was included that asked: “From the patient’s point of view, what are the top two most serious long-term effects the patient is currently experiencing?”, (hereafter referred to as the ‘long-term effects question’). The question was followed by a list of 15 possible long-term effects to help explore potential problems with the patients, but patients were also permitted to state problems that were not listed. The list of 15 problems was established collaboratively a priori among clinicians in the study as part of the development of the CRF and included ‘pain in the head and neck region’, ‘pain at flap donor site’, ‘difficulty swallowing/eating’, ‘difficulty breathing’, ‘hoarseness or difficulty speaking’, ‘dry mouth’, ‘numbness in the face and/or neck’, ‘neck stiffness’, ‘fatigue’, ‘social isolation’, ‘sialorrhea’, ‘trismus’, ‘osteonecrosis’, ‘fistula’, ‘dissatisfaction with facial appearance’, and ‘other’ followed by a free text field.

If the survivor agreed to participate, he/she was invited for a one-time assessment at the local clinic. The clinician recorded the patient’s response to the long-term effects question in the CRF. The data at each site were collected in paper form and then either scanned and sent electronically or sent in paper form by post to the coordinating center in Mainz, Germany, where it was entered into a web-based database created by the Evaluation Software Development company [17]; the only exceptions to this were the collaborators in Porto, Portugal, and Athens, Greece, who opted to enter the data into the database themselves.

The content of the free text field was examined manually, and new categories of side effects were created as required. This was done by first listing all the free text entries and then grouping similar entries together or assigning entries to one of the 15 options already listed.

Categorical data are presented as absolute values and percentages and continuous data as means with standard deviations and/or ranges. Logistic regressions were performed to explore characteristics associated with the top four most commonly reported problems adjusted for age as a categorical variable, diagnosis group, treatment group, gender, and UICC (Union for International Cancer Control) stage. Before the logistic regressions, the variables were examined for polychoric correlation to investigate multicollinearity as all variables in the model were ordinal or nominal. All confidence intervals (CI) reported are 95%. No ranking of the problems was considered as the survivors were not asked to indicate which of the two problems was the more serious or if they were equally serious. All analyses were carried out using SAS (Statistical Analysis Software) version 9.4.

## 3. Results

The EORTC 1629 project enrolled survivors at 26 sites in 11 countries; most countries were in Europe, but sites in Japan, Israel, and Brazil also participated. The first survivor was enrolled in October 2018 and the last in November 2021. Among the 1114 participants, 1033 responded to the long-term effects question. The majority were male and had oropharyngeal, oral cavity, or laryngeal cancer (Table 1). Most survivors had undergone multi-modal treatment, and in nearly 80% of the cases, the diagnosis was at least seven years in the past. The mean time since diagnosis was 9 years (range 5–36 years). The mean age at the time of the survey was 65.8 years (range: 23–93) and 97% of the survivors currently had no evidence of disease.

### 3.1. Frequency of Problems Reported

One hundred seventy-three survivors (17%) reported only one serious side effect, 770 (75%) reported two side effects, 32 (3%) reported three side effects, two (0.2%) reported four side effects, and 56 (5%) reported that they had no serious side effects. Although the survivors were only asked for two serious problems, a small portion wanted to state more than two, and this was permitted. ′Dry mouth′ was mentioned most often (*n* = 476, 46%), followed by ′difficulty swallowing/eating′ (*n* = 408, 40%), ′hoarseness/difficulty speaking′ (*n* = 169, 16%), and ′pain in the head and neck′ (*n* = 142, 14%) (Figure 1). A total of 328 survivors used the free text option to report additional problems. The free text responses resulted in the creation of nine new domains plus categories to capture situations where no details about the problem were given, if the patient indicated he/she had no serious problems at all, and a ′various other problems′ category for free text entries that did not fit well in any specific category. The nine new domains were: ′problems with taste and sense of smell′ (*n* = 40, 4%); ′dental problems′ (*n* = 28, 3%); ′coughing/sticky saliva′ (*n* = 25, 2%); ′problems with hearing (not tinnitus)′ (*n* = 19, 2%); ′tinnitus′ (*n* = 9, 1%); ′mental/memory problems′ (*n* = 13, 1%); ′nose problems (not trouble smelling)′ (*n* = 16, 2%); ′sexuality problems′ (*n* = 5, 0.5%); and ′arm/shoulder/hand problems′ (*n* = 13, 1%).

Four survivors (0.4%) indicated they had a serious problem but did not give details (‘problem not specified’ in Table 1). Sixty-two (6%) survivors indicated a problem that did not fit well into a specific category (‘various other problems’). This category covers a diverse mix of problems including, for example, tongue paralysis, dry eyes, watery eyes, Frey’s syndrome, fainting, dizziness, swelling, and muscle cramps. One survivor reported an inability to whistle as his most serious problem. Nose problems that were not related to smelling included bleeding and excessive mucous.

A small percentage of survivors reportedly had current evidence of disease (*n* = 32, 3%); these survivors were not excluded as recurrence or a second primary is also a reality for some patients who have survived more than five years since their first diagnosis. We looked at problems reported by these survivors on their own and found that two reported having no serious current problems. Their most frequent four problems were the same as the rest of the study population: 12 reported difficulty swallowing/eating, 11 had problems with pain, 9 with dry mouth, and 6 with hoarseness/difficulty speaking.

### 3.2. Associations of Four Most Frequently Mentioned Problems with Survivor Characteristics

Tests for polychoric correlation did not reveal evidence for substantial multicollinearity, with coefficients ranging from −0.12 to 0.4. The highest correlation was between treatment group and UICC stage. All odds ratios (ORs) reported in this section have been adjusted for age, gender, tumor subsite, UICC stage, and treatment received (Table 2).

A negative association with age was found for ‘pain in the head and neck’, with the two middle age groups (60–69 years and 70–79 years) compared to the survivors younger than 60 years old, with respective ORs of 0.6 (CI: 0.4–0.9) and 0.4 (CI: 0.2–0.6) (Table 2). This was an expected result and so out of interest we performed a small post hoc check to see whether there was a difference in the employment status among survivors reporting pain and those who did not. The percentages of survivors who were employed or self-employed and retired or unable to work were similar between the 142 survivors who reported pain as a serious problem and those who did not (employed or self-employed: 29% versus 24%; retired or unable to work: 59% versus 66%).

No strong evidence was found for associations between ‘difficulty swallowing/eating’ (OR: 1.1, CI: 0.9–1.5), ‘hoarseness/difficulty speaking’ (OR: 1.2, CI: 0.8–1.8), or ‘pain in the head and neck’ (OR: 0.8, CI: 0.5–1.2) and gender. However, some weak evidence for females being more likely than males to report ‘dry mouth’ as a serious problem was found (OR: 1.3, CI: 1.0–1.8)

In terms of tumor subsite groups, compared to oropharyngeal cancer survivors, survivors of hypopharyngeal (OR: 0.3, CI: 0.1–0.6), laryngeal (OR: 0.4, CI: 0.3–0.6), nasal cavity/paranasal sinus (OR: 0.3, CI: 0.1–0.7), and oral cavity cancer (OR: 0.6, CI: 0.4–0.9) were less likely to report ‘dry mouth’ as a serious problem. Survivors of nasal cavity/paranasal sinus cancer (OR: 0.4, CI: 0.2–0.9) and salivary gland cancer (OR: 0.4, 95% CI: 0.2–0.9) were also less likely to note eating/swallowing as a main problem compared to oropharyngeal cancer survivors. Also as compared to oropharyngeal cancer survivors, considerably increased odds were found for reporting ‘hoarseness/difficulty speaking’ among survivors of hypopharyngeal (OR: 5.4, CI: 2.5–11.5) and laryngeal cancer (OR: 7.2, CI: 4.3–12.3) (Table 2).

Survivors whose tumor was UICC stage II or IV at diagnosis had increased odds of reporting eating/swallowing problems compared to those with a stage I tumor (respectively, OR: 1.9, CI: 1.2–3.1; OR: 1.9, CI: 1.2–3.0). Survivors of stage III tumors also had increased odds, but the evidence of association was weaker (OR: 1.7, CI: 1.0–2.7). There was no evidence of an association between the UICC stage and the other three most frequent problems (Table 2).

Some evidence was found for an association between the type of treatment received and all of the four most frequently reported problems. Compared to survivors who had been treated with surgery alone, there were increased odds of reporting ‘dry mouth’ as a serious problem for survivors who had chemo-radiotherapy (CRT) (OR: 4.7, CI: 2.5–9.0), radiotherapy (RT) (OR: 2.9, CI: 1.5–5.5), and ‘surgery and RT +/− CT’ (OR: 3.2, CI: 1.8–5.6). Increased odds for ‘difficulty swallowing/eating’ were found for survivors who had CRT (OR: 2.1, CI: 1.1–3.9) or ‘surgery and RT +/− CT’ (OR: 1.8, CI: 1.1–3.2) compared to survivors with surgery only. All treatment groups had reduced odds of reporting ‘hoarseness/difficulty speaking’ as a serious problem compared to the surgery-only group. For ‘pain in the head and neck’, reduced odds were also found for survivors who had CRT (OR: 0.2, CI: 0.1–0.5) and RT (OR: 0.3, CI: 0.1–0.8).

## 4. Discussion

Our results showed that the most frequent serious long-term problems reported by HNC survivors were dry mouth, difficulty swallowing/eating, hoarseness/difficulty speaking, and pain in the head and neck, with a range of additional problems reported. In particular, we found evidence that oropharyngeal cancer survivors were more likely to report having a dry mouth as a main problem compared to nearly all other HNC sites. This is plausible as radiation doses to salivary glands are usually higher in the case of oropharyngeal cancer due to the vicinity of the oropharynx and involved nodal areas. Oropharyngeal cancer survivors also had increased adjusted odds of reporting difficulties in swallowing/eating as a main problem compared to nasal cavity/sinuses and salivary gland cancer survivors. The extent of these problems in oropharyngeal cancer patients treated with surgery or radiotherapy was investigated in a study from Denmark, reporting the highest symptom burden score for dry mouth at 12 months among radiation patients and sticky saliva scores for transoral surgery patients; however, the “problems with social eating scores” were not among the top problematic issues reported, but this study only included 44 patients and none of them were stage IV [18]. Among a Swedish population of long-term pharyngeal cancer survivors, dry mouth was the symptom with the most severe symptom burden [19].

Our results suggest that a higher UICC stage is associated with the odds of experiencing ‘difficulty swallowing/eating’ as a serious problem, which perhaps was detectable in our data because of the larger proportion of more severe tumor stages. Of course, patients with higher-stage tumors can expect to also receive more aggressive therapy, but our analysis controlled for treatment and still identified an association. It is also possible that survivors who had the advanced disease already had more swallowing and eating problems at diagnosis. Our study did not find an association between UICC stage and dry mouth, which was also reported by a study on oropharyngeal cancer survivors at a median of six years post-treatment [20]. Interestingly, the aforementioned study did find an association with gender, with females being more likely than men to have moderate/severe dry mouth. Our results showed weak evidence for this.

Our results for markedly increased odds of having difficulty speaking for laryngeal cancer survivors were not surprising given the core role of the larynx in speaking. Results from a prospective cohort study on laryngeal patients up to 1 year after laryngectomy showed the second highest symptom burden for trouble speaking at 1 year [8]. A long-term study on laryngeal patients reported that speech had improved between diagnosis and five years and had the fifth-highest mean symptom score after dry mouth, sexuality problems, sticky saliva, and coughing [9].

‘Pain in the head and neck’ showed some association with age in our study, with survivors younger than 60 being more likely than 60–69-year-olds or 70–79-year-olds to report pain as a serious problem. The extra look into the employment status of the survivors who reported pain versus those who did not revealed no notable difference in terms of distribution, so the current employment status does not seem to be an associated force behind this. A recent study from the United States reported an association between pain and adverse employment outcomes among cancer survivors, with survivors with moderate pain reporting early retirement and other reductions in employment more often [21]. However, only about 50% of that study population was at least five years’ post-treatment, and the study participants were asked about adverse employment events that had occurred at any time since their diagnosis.

The problems that were reported less frequently are nonetheless interesting and provided insight into problems that may not be captured on standard questionnaires. For example, the eye problems described by the survivors are not part of the European Organization for Treatment of Cancer (EORTC) core questionnaire or the head and neck module, and it is possible that issues related to survivorship are missing [14,15]. An extensive questionnaire specifically for cancer survivors recently completed Phase III testing and has moved on to Phase IV [22].

One of the limitations of these results is that we cannot say anything about the number of problems that the survivors would consider serious as we specifically asked for two problems and not a list of all problems they may be experiencing. The study may also have a selection bias because the data collection was performed in the hospital, meaning that the survivors who had the most serious problems of all may not have been able to participate. Moreover, because we asked for the survivors’ most serious problems, it is possible that the problem is not particularly detrimental to them if the survivor, in general, is doing well but wanted to nonetheless provide an answer to what for them is the most serious problem. If survivors who truly did not have a serious problem felt compelled to nonetheless give an answer, the results may have been inflated. However, the aim of the analysis was not to document symptoms that are ‘serious’ as measured on an objective scale, but rather are considered serious by the patient, regardless of the extent, so this would have been difficult to mitigate. This reflects a fundamental difference between asking a patient to report the problems they feel are serious and systematically measuring a set of symptoms; what one patient considers serious may not be serious to the next. We also cannot rule out that additional health problems or current irritations had a greater influence on the responses than the effect of the HNC disease and treatment. In addition, while some clinicians noted in the free text field when a survivor reported having no problems, this was not specifically asked for, and it could be that some of the survivors who appeared to not have responded to the question actually did not have any problems to report; however, we cannot know this. The list of potential problems provided in the CRF may have steered the survivors towards specifically these problems; nevertheless, a framework for the discussion was wanted and at least with the list all clinicians had the same starting point. It is possible that the face-to-face setting had an influence on the patients’ responses, and that they may have been more forthcoming about more sensitive issues such as problems with sexuality in a written format. The strengths of this analysis include the large number of survivors included and the international setting with a range of diagnoses, tumor stages, and treatments. In addition, this study’s requirement to be at least five years’ post-diagnosis provides a more accurate picture of the main serious problems experienced by survivors in the long term compared to studies including up to 1 or 3 years of follow-up.

The importance of long-term follow-up care and long-term assessment of problems is part of the recently published survivorship care recommendations of the European Head and Neck Society [23]. We have identified a range of long-term effects in an international population of long-term HNC survivors, with dry mouth, difficulties in eating, swallowing, and speaking, and pain as the most frequent serious problems. Information on the differences in these problems by tumor subsite and treatment group may be useful as examples of what newly diagnosed patients may expect. Future research could focus on how best to intervene to alleviate these problems.

## Figures and Tables

**Figure 1 healthcare-11-00906-f001:**
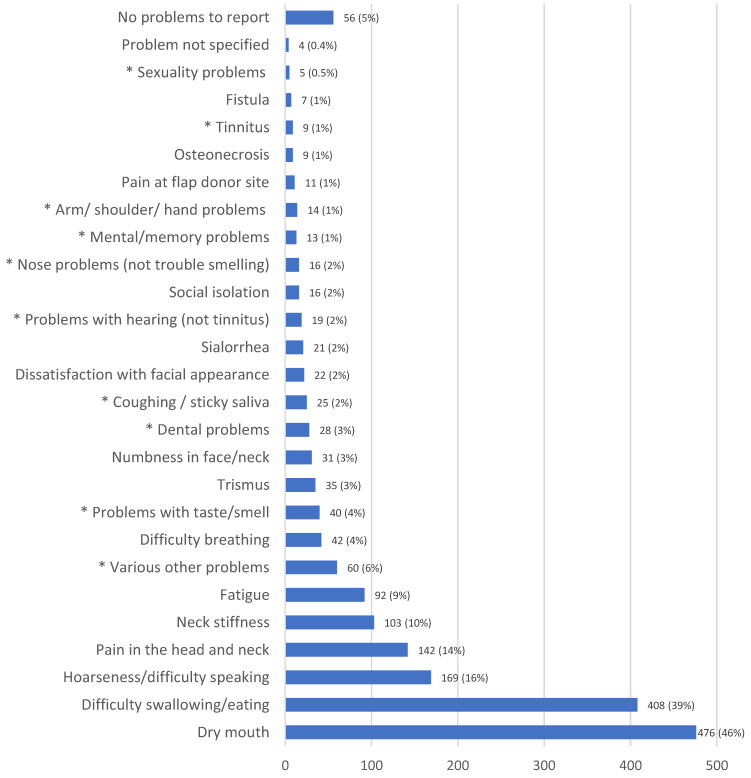
Frequency and percentages of the most serious side effects reported among head and neck cancer survivors. Percentages indicate the proportion of patients who reported the problem. The * symbol indicates a category that resulted from the free text field analysis of the ‘other’ category on the Case Report Form.

**Table 1 healthcare-11-00906-t001:** Characteristics of the 1033 survivors who gave a response to the long-term effects question.

	*n*	%
Gender		
Male	727	70%
Female	306	30%
Age in years		
<60	246	24%
60–69	423	41%
70–79	275	27%
80+	89	9%
Geographic area		
Northern Europe	237	23%
Central/Western Europe	440	43%
Southern Europe	218	21%
Israel	10	1%
Japan	30	3%
Brazil	98	9%
Tumor sub-site		
Oropharynx	366	35%
Oral cavity	221	21%
Larynx	188	18%
Nasopharynx	79	8%
Salivary gland	53	5%
Hypopharynx	46	4%
Nasal cavity and sinuses	34	3%
Unknown primary	46	4%
Treatment		
Surgery	112	11%
RT	118	11%
CRT	302	29%
RT +/− CT and ND	103	10%
Surgery and RT +/− CT	397	38%
Unknown	1	0%
UICC stage (Version 7)		
I	195	19%
II	160	15%
III	234	23%
IV	416	40%
Missing/Unknown	28	3%
Time since diagnosis (years)		
5 to 6	219	21%
7 to 8	367	36%
9 to 10	204	20%
>10	243	24%

RT: radiotherapy; CRT: chemo-radiotherapy; CT: chemotherapy; ND: neck dissection. UICC: Union for International Cancer Control. Note: “Surgery and RT +/− CT” includes all surgeries that were not neck dissections; not all categories sum to exactly 100% due to rounding.

**Table 2 healthcare-11-00906-t002:** Adjusted odds ratios and 95% confidence intervals for issues reported as being a serious long-term effect by head and neck cancer survivors.

	Dry Mouth		Difficulty Swallowing/Eating		Hoarseness/ Difficulty Speaking		Pain in the Head and Neck	
	OR	95% CI	OR	95% CI	OR	95% CI	OR	95% CI
Gender (reference = male)								
female	1.3	(1.0–1.8)	1.1	(0.9–1.5)	1.2	(0.8–1.8)	0.8	(0.5–1.2)
Age (reference = less than 60 years old)								
60–69 years	1.1	(0.8–1.6)	1.1	(0.8–1.6)	0.9	(0.6–1.5)	**0.6**	**(0.4–0.9)**
70–79 years	1.0	(0.7–1.5)	1.5	1.0–2.3)	1.4	(0.8–2.3)	**0.4**	**(0.2–0.6)**
80 years or older	0.7	(0.4–1.1)	1.0	(0.6–1.7)	0.7	(0.3–1.5)	0.5	(0.2–1.1)
Tumor subsite (reference = oropharynx)								
hypopharynx	**0.3**	**(0.1–0.6)**	1.9	(1.0–3.7)	**5.4**	**(2.5–11.5)**	0.8	(0.2–2.3)
larynx	**0.4**	**(0.3–0.6)**	0.9	(0.6–1.4)	**7.2**	**(4.3–12.3)**	0.5	(0.2–1.0)
nasal cavity and sinuses	**0.3**	**(0.1–0.7)**	**0.4**	**(0.2–0.9)**	0.5	(0.1–2.4)	0.6	(0.2–2.1)
nasopharynx	0.9	(0.5–1.5)	0.6	(0.3–1.0)	0.5	0.2–1.9)	2.3	(1.0–5.1)
oral cavity	**0.6**	**(0.4–0.9)**	0.9	(0.6–1.4)	1.5	(0.8–2.7)	1.1	(0.6–1.9)
salivary gland	0.6	(0.3–1.1)	**0.4**	**(0.2–0.9)**	0.6	(0.2–1.9)	1.2	(0.6–2.7)
unknown primary	1.2	(0.6–2.4)	1.0	(0.5–2.0)	0.8	(0.2–3.8)	0.6	(0.2–1.8)
UICC stage (reference = UICC I)								
UICC stage II	1.2	(0.7–1.9)	**1.9**	**(1.2–3.1)**	0.9	(0.5–1.7)	1.4	(0.7–2.6)
UICC stage III	0.8	(0.5–1.3)	1.7	(1.0–2.7)	1.8	(0.9–3.4)	1.0	(0.5–2.0)
UICC stage IV	0.9	(0.6–1.4)	**1.9**	**(1.2–3.0)**	1.0	(0.6–1.9)	1.2	(0.6–2.2)
Treatment group (reference = surgery only)								
CRT	**4.7**	**(2.5–9.0)**	**2.1**	**(1.1–3.9)**	**0.3**	**(0.1–0.6)**	**0.2**	**(0.1–0.5)**
RT	**2.9**	**(1.5–5.5)**	1.2	(0.6–2.3)	**0.3**	**(0.1–0.6)**	**0.3**	**(0.1–0.8)**
RT +/− CT and ND	2.0	(1.0–4.0)	1.9	(0.9–3.8)	**0.2**	**(0.1–0.6)**	1.0	(0.4–2.4)
Surgery and RT +/− CT	**3.2**	**(1.8–5.6)**	**1.8**	**(1.1–3.2)**	**0.5**	**(0.2–0.9)**	0.8	(0.4–1.4)

RT: radiotherapy; CRT: chemo-radiotherapy; CT: chemotherapy; ND: neck dissection. OR: odds ratio (mutually adjusted for the other predictors); CI: confidence interval; UICC: Union for International Cancer Control. Bolded entries show evidence of a statistical difference.

## Data Availability

The data are not publicly available until the main articles stemming from the EORTC 1619 project have been published. After that, data may be requested from the data repository of the EORTC.

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
