# Peer review of "Serious Long-Term Effects of Head and Neck Cancer from the Survivors’ Point of View"

_healthcare, 2023, doi:10.3390/healthcare11060906_

Round 1

Reviewer 1 Report

The authors have conducted a rather large cross-sectional study in a population of patients who suffer serious long-term effects of cancer and cancer treatment. 

Abstract: please state a rationale for this study. 

Introduction:
- the first sentence is very long and therefore difficult to read; please add a percentage for the share of HNC patients. 
- The authors state their rationale as: 'An aspect that is missing from this type of assessment is the patients’ opinion on what is the most serious side-effect he or she is currently suffering from.' Although that is true, and I can think of reasons why this is important, the authors do not state why this is an important knowledge gap to study. 

Methods:
- 'The list of 15 problems was established collaboratively a priori among clinicians in the study as part of the development of the CRF and included ‘pain in the head and neck region’, ‘pain at flap donor site’, ‘difficulty swallowing/eating’, ‘difficulty breathing’, ‘hoarseness or difficulty speaking’, ‘dry mouth’, ‘numbness in the face and/or neck’, ‘neck stiffness’, ‘fatigue’, ‘social isolation’, ‘sialorrhea’, ‘trismus’, ‘osteonecrosis’, ‘fistula’, ‘dissatisfaction with facial appearance’, and ‘other’ followed by a free text field.' Why were there no patients involved in creating this list?
- 'The content of the free text field was examined manually and new categories of side effects were created as required. This was done by first listing all the free-text entries and then grouping similar entries together or assigning entries to one of the 15 options already listed.' Did only one author look at this, or was there some sort of validation by one or more of the other authors?

Results:
- if the question was about the top two most serious side effects, how can it be that 32 (3%) reported three side effects, and two (0.2%) reported four side effects?
- Figure 1 would benefit from presenting percentages as well.
- 'No strong evidence for associations between any of the four most frequently reported problems and gender were found, ...' Please provide OR and CI. 

Discussion:
- As I understand, the physician read the question about the top two most serious long-term effects to the patient, and did the patient then state an answer after the physician read all 15 options. Do the authors think they would have gotten different responses if the question was provided through a survey? I think this might have an effect on the answers that were collected.
- Ideally, the logistic regression would have been adjusted for educational level and marital status. I assume these data were not available?
- The authors state that 'This reflects a fundamental difference between asking a patient to report the problems they feel are serious and systematically measuring a set of symptoms; what one patient considers serious may not be serious to the next.' This is a serious limitation, and I feel the focus of the discussion should not solely be why certain side effects were reported, but if the data is representative.  

Overall: perhaps it is style, but shorter sentences would increase the readability of this paper. The paper would benefit from English language editing. 

Reviewer 2 Report

This article described long term effects of head and neck cancer and attempts to identify factors associated with specific groups of symptoms. While the study group is large and the scientific question is of great relevance both to cancer survivors and clinicians, I have several reservations. 

1. Selection bias should be discussed as a potential limitation, since the survivors that did not agree to participate and come for the assessment at their local clinic could potentially be ones most (or least) burdened with long-term cancer effects

2. The numbers in table 1 do not sum up to 100% (101% for age groups, 99% for treatment groups). Age and time since diagnosis would also best be presented and analyzed as continuous variables instead of categories

3. The authors state that 32 survivors reported three serious side effects and two reported four, at the same time saying that the question asked concerned specifically two most serious long-term effects. Were the patients allowed to report more than two symptoms, then?

4. In figure 1, percentages instead of or at least alongside numbers would be useful

5. How did the authors establish reference categories for tumor entity (oropharynx) and treatment group (surgery alone)? Was it not of interest how survivors of larynx cancer compare, e.g., to nasopharynx cancer survivors, or how RT alone compares to RT alone in terms of long term effects?

6. Again, age should be analyzed as continous instead of ordinal variable when included in the LR models. 

7. Results for association with employment status are first reported in the discussion, while they should be reported in the results section.

Reviewer 3 Report

Dear editor and dear authors,

Thank you for the opportunity to review your paper entitled "Serious long-term effects of Head and Neck Cancer From the Survivors' point of view."

I appreciate the paper. The introduction and discussion are well-grounded from a theoretical point of view. The methodology and results are clear and objective, and the tables presented help the reader understand the study.

It identifies in the conclusions the future impact of this study and its contributions.

How the entire investigation was described is attractive to the reader.

If I am bold, this is a suggestion for a future discussion, introducing the country variable in the data analysis. For example, the incidence of nasopharynx cancer is different in Europe and Japan, as well as the therapeutic approach. So, it will be interesting to compare the long terms effects.

A small note to correct page 9, last line, "a survivors".

Thank You, 

Congratulations on your work.

Round 2

Reviewer 1 Report

The authors have improved their manuscript sufficiently.

I still believe the research design could have been improved by including patients in the creation of the symptom list, review by a second author for the categorizing the results from the free text fields, and anticipating on certain types of bias. However, the authors explain sufficiently why certain choices were made or which bias could have been introduced by the study design.